# Discovery of Novel Cinnamide Fungicidal Leads with Optical Hydroxyl Side Chain

**DOI:** 10.3390/molecules27165259

**Published:** 2022-08-17

**Authors:** Weiwei Wang, Jiazhen Jiang, Zhenhua Zhang, Mingan Wang

**Affiliations:** Innovation Center of Pesticide Research, Department of Applied Chemistry, China Agricultural University, Beijing 100193, China

**Keywords:** cinnamamide fungicide, 3-aryl-7-methyl-6,7-dihydroxyoct-2-enamide, 3-aryl-7-methyl oct-2,6-dienamide, asymmetric dihydroxylation, absolute configuration, fungicidal activity

## Abstract

In order to overcome the resistance of phytopathogens to commercial fungicides, a series of optical 2-methyl-2,3-diol-5-pentyl-based cinnamamide derivatives were rationally designed, synthesized, characterized, and evaluated for their in vitro and in vivo fungicidal activities. The bioassay results indicated that the EC_50_ (concentration for 50% of maximal effect) values of (*R*)-**11f**, (*R*)-**11m**, (***S***)-**11m** and (*R*)-**11n** were 0.16, 0.28, 0.41 and 0.47 µg/mL in the in vitro evaluation against *Sclerotinia sclerotiorum*, respectively, while compounds (*R*)- and (***S***)-**11i**, (*R*)- and (***S***)-**11j** exhibited excellent in vivo fungicidal activity against *Pseudoperonspera cubensis* with inhibition rates of 100% at 400 μg/mL. These findings supported the idea that optical 2-methyl-2,3-diol-5-pentyl-containing cinnamamides (*R*)- and (***S***)-**11i**, (*R*)- and (***S***)-**11j** with 2-chloro-4-trifluoromethyl aniline and 2-(4-chlorophenyl) aniline showed excellent in vivo fungicidal activity against *S. sclerotiorum* and *P. cubensis* and were promising fungicide candidates.

## 1. Introduction

Dimethomorph, flumorph and pyrimorph are widely used excellent amide fungicides in agriculture fields (Figure 1). However, resistance of phytopathogens to them has developed due to their wide application and their similar structures [1,2,3,4,5,6,7]. The resistance mechanisms have been confirmed to relate with point mutation in cellulose synthase 3 (CesA3) [1,4]. In order to overcome this problem, some works addressing this issue have been published, and several compounds had good fungicidal activities against the tested phytopathogens [8,9,10], but all of these molecules retained two (hetero)aryl groups such as benzene, pyridine and isothiazole. For example, isothiamorph was found to exhibit excellent in vivo fungicidal activity against *Pseudoperonspera cubensis* with both fungicidal activity and systemic acquired resistance [10]. In practice, how to find the novel chemical structures to overcome resistances is difficult and a challenge for agricultural chemists, and the costs of developments are high. To address this issue, our initial strategy was to replace one of the aryl groups in the molecules of dimethomorph, flumorph and pyrimorph with non-aryl groups, and change the morpholine motif into the other amines. However, we faced the question of how to find a suitable functional group.

3,7-Dimethyl-7-hydroxy-2-octen-6-olide **1** (Figure 2) is a naturally occurring seven-membered lactone that was isolated from the honeybee fungal entomopathogen *Ascosphaera apis*, as well as the fruit of plant *Litsea cubeba* in Tibet, and it exhibited good antifungal and antioxidant activities [11,12]. As reported in the literature, the lactone motif plays an important role in the chemical communication between wide varieties of organisms [13], and this class of naturally occurring lactones was confirmed to have a wide range of biological properties such as antifungal, antimicrobial, and phytotoxic activities as well as cytotoxicity against human tumor cells [14,15,16,17]. We paid attention to the total synthesis of natural products with seven-membered lactone moieties and their biological activities in previous papers [18,19,20,21,22]; the synthesis and biological activity evaluation of the racemic 3,7-dimethyl-7-hydroxy-2-octen-6-olide (**1**), 3,7-dimethyl-2,6- octadien-1,6-olide, 3-aryl-7-methyl-7-hydroxy-2-octen-6-olide, and their 3-(2-hydroxy- propan-2-yl)-4,5-dihydrobenzo[c]oxepin-1-(3*H*)-one analogues were carried out in our laboratory [18,19,20,21,22].

Among them, 3-phenyl-7-methyl-7-hydroxy-2-octen-6-olide (**2**) was found to exhibit more excellent fungicidal activities against several phytopathogens than naturally occurring 3,7-dimethyl-7-hydroxy-2-octen-6-olide (**1**) and the other derivatives, which indicated that the C_3_-aryl significantly improved the fungicidal activities of this type of seven-membered lactone (Figure 2) [22]. The four isomers of 6,7-dihydroxy-3,7- dimethyloct-2-enoic acid (**3**) were also synthesized, and we found that the chiral acid (***Z***, ***S***)-isomer-**3** was a good lead compound with excellent in vivo antifungal activities against several plant pathogens in our previous report (Figure 3) [23]. In the other aspect, (***Z***/***E***)-3,7-dimethylocta-2,6-dienamides (**4**), their 6,7-epoxy analogues (**5**) and optical (6***R*** or 6***S***)-3,7-dimethyl-6,7-dihydroxyoct-2-enamides (**6**) were found to exhibit in vitro and in vivo fungicidal activities against several phytopathogens in our previous reports (Figure 3), but the 6,7-epoxy analogues (**5**) decreased the fungicidal activities in comparison with the amides **4** and **6** [24,25].

Considering the above results, our strategy was that one type of novel amide (**11a**–**11p**) was designed when CH_3_ of amides **6** was replaced with the aryl groups, or one aryl of dimethomorph, flumorph and pyrimorph was replaced with the optical hydroxyl side chain in ***Z***-**3** (Figure 4).

These structures were different from the dimethomorph, flumorph and pyrimorph, as they could not only improve the in vitro and in vivo fungicidal activities against phytopathogens, but also could overcome the resistance issue. The synthetic route is shown in Figure 5, and the fungicidal activity evaluation is reported in this article.

## 2. Results and Discussion

### 2.1. Chemistry

As indicated in a previous report, lactone **2** was designed [26]. The lactone **2** and analogues were synthesized and evaluated for their fungicidal activities [22]. It was found that (***R***)-**2** was the most active compound with EC_50_ values in the range of 0.2–13.5 µg/mL against the tested phytopathgens, better than its (***S***)-isomer and racemic mixture. The scanning electron microscope (SEM) and transmission electron microscope (TEM) observations indicated that compounds (***S***)-**2** had a significant impact on the structure and function of the hyphal cell wall of *S. sclerotiorum* mycelium [22]. With comparison of those data with that of naturally occurring (***R***)-**1**, it was found that the C_3_-aryl significantly improved the fungicidal activities of this type of seven-membered lactone [22]. The amides **4**, **5**, and **6** having C_3_-CH_3_ were also synthesized and evaluated for their fungicidal activities. Some of them exhibited in vitro fungicidal activities against the tested phytopathogens, but were much weaker than those of (***R***)-**1**, pyrimorph and dimethomorph, while several compounds showed in vivo fungicidal activities against *P**. cubensis* and *Erysiphe graminis*, but which were also weaker than that of the chiral acid (***Z***, ***R***)-**2** and (***Z***, ***S***)-**2** [24,25]. Therefore, we attempted to replace the C_3_-CH_3_ with similar aryl groups as in the previous report [22], and hope to improve the in vitro and in vivo fungicidal activities; thus, the amides **11a**–**11p** were designed, synthesized and evaluated for their fungicidal activities in this article. 

The olefin acids **7** and **8a**–**8d** were prepared following the procedures in the previous report [26]. The olefin acids **8a**–**8d** could easily react with morpholine and (4-(*tert*-butyl) phenyl)methanamine using 1-ethyl-3-(3-dimethylaminopropyl)carbodiimide hydrochloride (EDCI) and N-hydroxybenzotrizole (HOBT) as the catalysts to provide the amides **10a**–**10h**, but the 2-chloro-4-(trifluoromethyl)aniline and 2-(4-chlorophenyl) aniline could not take part in the reaction because of the weak nucleophilic reactivity and hindrance. Then, the olefin acid **7** was initially transferred into acid chloride, and the acid chloride reacted smoothly with 2-chloro-4-(trifluoromethyl)aniline and 2-(4-chloro phenyl)aniline to afford the amides **9a** and **9b**. The amides **9a** and **9b** took part in the stereoselective Mizoroki–Heck arylation with 4-(*tert*-butyl)iodobenzene, 4-phenyl-iodobenzene, 1-iodonaphthalene and 2-iodonaphthalene to give the amides **10i**–**10p** according to the protocol in the previous report [26]. Finally, we carried out the regioselective Sharpless asymmetric dihydroxylation of the amides **10a**–**10h** at remote C=C double bond with AD-mix-β and AD-mix-α to produce the chiral amides (***R***)- and (***S***)-**11a**–**11p** in 75–98% high yields and 90.4–99.4% high ee values as in the previous paper [21,22,23]. 

### 2.2. The In Vitro and In Vivo Fungicidal Activities

After completion of synthesis, the in vitro and in vivo fungicidal activities of compounds **11a**–**11p** were evaluated, as shown in Table 1, Table 2 and Table 3. The data in Table 1 indicate that all of compounds ((***R***)-**11a**–**11p** and (***S***)-**11a**–**11p**) with dihydroxyl had weak fungicidal activities against *A. solani*, *P. capsici*, *B. cinerea and R. solani*, while some of them (eg. (***R***)**-11f** and (***R***)**-11n**) had excellent fungicidal activities against *S. sclerotiorum*. The EC_50_ values of these compounds having strong fungicidal activities against *S. sclerotiorum* were determined and provided in Table 2. These data indicated that (***R***)- and (***S***)-**11a**, **11c**, **11e** and **11g** almost lost their fungicidal activities against *S. sclerotiorum* after dihydroxylation, however (***R***)- and (***S***)-**11b**, (***R***)- and (***S***)-**11f**, and (***R***)-**11h** had good fungicidal activities with EC_50_ values of 0.16–67.8 µg/mL against *S. sclerotiorum* after dihydroxylation. These compounds ((***R***)- and (***S***)-**11m**, **11n**, **11o**) with 2-chloro-4-trifluoromethylaniline and 2-(4-chlorophenyl)aniline exhibited excellent fungicidal activities against *S. sclerotiorum* with EC_50_ values of 0.28–11.4 µg/mL, which indicated that the dihydroxyl groups significantly improved their in vitro fungicidal activities. To our surprise, compounds (***R***)- and (***S***)-**11****e** (***R***_1_ + ***R***_2_ = morpholino) had very weak in vitro fungicidal activities against five phytopathagens, so we primarily deduced that the α-naphthyl group had a bigger hindrance than the 4-*t**ert*-butyl-phenyl, 4-phenyl-phenyl and β-naphthyl group, as they cannot enter the active site of the target. All the data in Table 2 showed that the ***R***-configuration is much better than the ***S***-configuration for in vitro fungicidal activities; the chiral amides have much better in vitro fungicidal activities than the seven-membered lactones such as **2**, (***R***)- and (***S***)-**2** [22]. Among these compounds, the EC_50_ values of (***R***)-**11f**, (***R***)-**11m**, (***S***)-**11m** and (***R***)-**11n** were 0.16, 0.28, 0.41 and 0.47 µg/mL against *S. sclerotiorum*, respectively. They exhibited the best in vitro fungicidal activities in comparison with the chiral lactone lead (***R***)-**2**, (***S***)-**2 [22]** and the chiral amides **6** [25]. 

In order to confirm their fungicidal activities, the in vivo fungicidal activities of compounds **11a**–**11p** were assessed, and the results are provided in Table 3. For (***R***)- and (***S***)-**11m**, **11n**, they only showed 50–60% efficacy against *P. cubensis*, weaker than that of positive control flumorph, pyrimorph, and the lead (***R***)-**3** and (***S***)-**3**. To our surprise, compounds (***R***)- and (***S***)-**11i** and **11j**, with weak in vitro fungicidal activities, exhibited excellent in vivo fungicidal activities with 100% efficacies at 400 µg/mL, better than that of the positive control flumorph. They still had 20–98% efficacies when concentration decreased to 100 µg/mL, much better than the chiral acid leads (***R***)-**3** and (***S***)-**3**. Notably, (***R***)- and (***S***)-**11j** remained at 5% and 10% efficacies when concentration decreased to 6.25 µg/mL. These results showed that (***R***)- and (***S***)- **11i**, **11j**, **11m** and **11n** were excellent lead compounds worthy of further optimization. This work is currently under way in our group. 

## 3. Experimental Procedures

### 3.1. General Information

All reactions were performed with magnetic stirring. Unless otherwise stated, all reagents were purchased from commercial suppliers (Energy Chemical, Shanghai, China) and used without further purification. Organic solutions were concentrated under reduced pressure using a rotary evaporator or oil pump. Flash column chromatography was performed using Qingdao Haiyang silica gel (200–300 mesh). Melting points were measured on a Yanagimoto apparatus (Yanagimoto MFG Co., Kyoto, Japan) and are uncorrected. ^1^H and ^13^C NMR spectra were obtained on Bruker DPX 300 spectrometer (Bruker Biospin Co., Stuttgart, Germany) with CDCl_3_ as a solvent and TMS as an internal standard; chemical shifts were presented with δ. HR-ESI-MS spectra were analyzed on Bruker Apex II mass spectrometer (Bruker Co., Bremen, Germany). The solvents were analytical grade and newly distilled before usage. The e.e values were analyzed by an Agilent LC 1100 HPLC instrument equipped with a chiral Chiralpak AD column (250 mm × 4.6 mm), eluent: hexane/isopropanol (95:5; 90:10; 85:15), flow rate: 1.0 mL/min, UV detection wavelength: 230 nm. (See Appendix A).

### 3.2. Synthesis of the Olefin Acids ***7*** and ***8a***–***8d***

The olefin acid **7** was prepared through 5-step reactions using 2-methylbut-3-en-2-ol as the starting material following the procedures. The olefin acids **8a**–**8d** were prepared through the stereoselective Mizoroki–Heck arylation of **7** with 4-(*tert*-butyl)-iodobenzene, 4-phenyl-iodobenzene, 1-iodonaphthalene and 2-iodonaphthalene according to the protocol in the previous reports, and their spectral data were identical with that reported in the literature [22,26].

### 3.3. Synthesis of the Amides ***9a*** and ***9b***

Synthesis of the amides **9a** and **9b**: The olefin acid **7** (1.0 g, 6.5 mmol) and 100 mL CH_2_Cl_2_ were added into a 250 mL single-necked flask in an ice-water bath, then we added 1 mL oxalyl dichloride and 3 drops of DMF in a stirred condition. After the bubble disappeared, we removed the ice-water bath, and reacted 1–2 h. The solvent was removed in vacuo to afford the acid chloride. The acid chloride CH_2_Cl_2_ (10 mL) solution and pyridine (1 mL) were added dropwise into the 20 mL CH_2_Cl_2_ solution of 2-chloro-4-(trifluoromethyl)- aniline (2.00 g, 10.2 mmol) or 2-(4-chlorophenyl) aniline (2.07 g, 10.2 mmol) at the ambient temperature and stirred for 8–10 h. After the reaction was completed, 30 mL water was added into the mixture, poured into the separatory funnel, shaken and separated into the organic phase. Then, the water phase was extracted with CH_2_Cl_2_ (3 × 30 mL), combined with the organic phase, and the organic phase was dried over anhydrous Na_2_SO_4_. The solvent was removed in vacuo, and the residue was recrystallized using petroleum ether to give white solid **9a** or **9b**.

(*E*)-*N*-(2-Chloro-4-(trifluoromethyl)phenyl)-7-methylocta-2,6-dienamide **9a****.** A white solid, yield 42%, m.p. 88–90 °C. ^1^H NMR (300 MHz, CDCl_3_) δ: 8.68 (d, J = 8.7 Hz, 1H), 7.76 (s, 1H), 7.64 (s, 1H), 7.54 (d, J = 8.7 Hz, 1H), 7.05 (dt, J = 15.2, 7.5 Hz, 1H), 6.02 (d, J = 15.2 Hz, 1H), 5.15–5.10 (m, 1H), 2.35–2.27 (m, 2H), 2.23–2.15 (m, 2H), 1.71 (s, 3H), 1.63 (s, 3H). HR-ESI-MS, *m/z*: C_16_H_18_ClF_3_NO [M+H]^+^, Cacld. 332.1024, Found: 332.1028.

(*E*)-*N*-(2-(4-Chlorophenyl)phenyl)-7-methylocta-2,6-dienamide **9b****.** A white solid, yield 45%, m.p. 105–107 °C. ^1^H NMR (300 MHz, CDCl_3_) δ: 8.34 (brs, 1H), 7.47 (d, J = 8.4 Hz, 2H), 7.42–7.30 (m, 3H), 7.24–7.11 (m, 2H), 6.99 (s, 1H), 6.90 (dt, J = 15.8, 7.5 Hz, 1H), 5.70 (d, J = 15.8 Hz, 1H), 5.12–5.06 (m, 1H), 2.35–2.28 (m, 2H), 2.23–2.15 (m, 2H), 1.68 (s, 3H), 1.59 (s, 3H). HR-ESI-MS, *m/z*: C_21_H_23_ClNO [M+H]^+^, Cacld. 340.1463, Found: 340.1461.

### 3.4. Synthesis of the Amides ***10a***–***10p***

The general synthetic method (A): 50 mL CH_2_Cl_2_ and olefin acid **8a** (686 mg, 2.4 mmol) were added into a 250 mL single-necked flask, then EDCI (652 mg, 3.4 mmol) and HOBt (458 mg, 3.4 mmol) were added into the mixture and stirred. After the mixture was clear, morpholine (0.5 mL, 5.75 mmol) was added and reacted for 10 h. Then, 30 mL water was added into the mixture, poured into the separatory funnel, shaken and separated into the organic phase. Then the water phase was extracted with CH_2_Cl_2_ (3 × 30 mL) and the organic phase was combined and dried over anhydrous Na_2_SO_4_. The solvent was removed in vacuo, and the residue was subjected to a flash silica gel chromatography and washed with petroleum ether/EtOAc (*v*:*v* = 3:1) to give a colorless liquid **10a**. Compounds **10b**–**10h** were prepared in a similar way.

The general synthetic method (B): To add **9a** (400 mg, 1.20 mmol), Pd(OAc)_2_ (14 mg, 0.06 mmol), P(*o*-MeC_6_H_4_)_3_ (42 mg, 0.14 mmol), 4-phenyl-iodobenzene (640 mg, 2.29 mmol), and 6 mL N(C_2_H_5_)_3_ into a 25 mL three-necked flask under N_2_ atmosphere. The mixture was stirred and heated to 110 °C for 20 h, then cooled down to the room temperature; we adjusted pH to 2 using 1M HCl solution and added 30 mL water. The water phase was extracted with EtOAc (30 mL × 3), the organic phase was combined and dried over anhydrous Na_2_SO_4_. The solvent was removed under vacuum, and the residue was subjected to a flash silica gel chromatography and washed with petroleum ether/EtOAc (*v*:*v* = 100:1) to afford a white solid **10i**. Compounds **10j**–**10p** were prepared in a similar approach. 

(2*E*)-3-(4-*tert*-Butylphenyl)-7-methyl-1-morpholinoocta-2,6-dien-1-one **10a**. A colorless liquid, 307 mg, yield 71%. ^1^H NMR (300 MHz, CDCl_3_) δ: 7.41–7.31 (m, 4H), 6.12 (s, 1H), 5.11 (t, *J* = 7.2 Hz, 1H), 3.75–3.49 (m, 8H), 2.72 (t, *J* = 7.8 Hz, 2H), 2.14–2.05 (m, 2H), 1.64 (s, 3H), 1.50 (s, 3H), 1.33 (s, 9H). ^13^C NMR (75 MHz, CDCl_3_) δ: 166.75, 151.05, 150.73, 137.45, 131.73, 125.86, 125.03, 123.34, 118.70, 66.53, 46.43, 41.40, 34.24, 31.15, 30.93, 27.01, 25.32, 17.33. HR-MS (ESI) *m/z*: Calcd. for C_23_H_33_NO_2_ [M+H]^+^, 356.2584; Found, 356.2586.

(2*E*)-*N*-(4-(*tert*-Butyl)benzyl)-3-(4-(*tert*-butylphenyl)-7-methylocta-2,6-dienamide **10b**. A colorless solid, 364 mg, yield 69%, m.p. 99–100 °C. ^1^H NMR (300 MHz, CDCl_3_) δ: 7.38 (d, *J* = 8.4 Hz, 2H), 7.36 (s, 4H), 7.28 (d, *J* = 8.4 Hz, 2H), 6.02 (s, 1H), 5.97 (s, 1H), 5.17 (t, *J* = 7.2 Hz, 1H), 4.50 (d, *J* = 5.6 Hz, 2H), 3.14 (t, *J* = 7.8 Hz, 2H), 2.20–2.10 (m, 2H), 1.63 (s, 3H), 1.52 (s, 3H), 1.35 (s, 9H), 1.34 (s, 9H). ^13^C NMR (75 MHz, CDCl_3_) δ: 166.18, 154.46, 151.17, 150.11, 138.22, 135.19, 131.73, 127.44, 125.99, 125.27, 125.00, 123.62, 119.51, 42.90, 34.25, 34.18, 31.03, 30.96, 30.39, 27.28, 25.30, 17.36. HR-MS (ESI) *m/z*: Calcd. for C_30_H_41_NO [M+H]^+^, 432.3261; Found, 432.3255.

(2*E*)-3-(4-Phenylphenyl)-7-methyl-1-morpholinoocta-2,6-dien-1-one **10c**. A colorless liquid, 225 mg, yield 73%. ^1^H NMR (300 MHz, CDCl_3_) δ: 7.61 (dd, *J* = 7.9, 2.7 Hz, 4H), 7.53–7.41 (m, 4H), 7.36 (t, *J* = 7.2 Hz, 1H), 6.19 (s, 1H), 5.13 (t, *J* = 7.0 Hz, 1H), 3.77–3.53 (m, 8H), 2.80 (t, *J* = 8.1 Hz, 2H), 2.08–2.17 (m, 2H), 1.65 (s, 3H), 1.51 (s, 3H). ^13^C NMR (75 MHz, CDCl_3_) δ: 166.59, 150.41, 140.75, 140.11, 139.38, 131.89, 128.49, 127.17, 126.81, 126.65, 123.22, 119.38, 66.57, 46.46, 41.44, 31.18, 26.94, 25.33, 17.35. HR-MS (ESI) *m/z*: Calcd. for C_25_H_30_NO_2_ [M+H]^+^, 376.2271; Found, 376.2270.

(2*E*)-*N*-(4-(*tert*-Butyl)benzyl)-3-(4-phenylphenyl)-7-methylocta-2,6-dienamide **10d**. A colorless solid, 330 mg, yield 70%, m.p. 109–111 °C. ^1^H NMR (300 MHz, CDCl_3_) δ: 7.63–7.57 (m, 4H), 7.49–7.43 (m, 4H), 7.42–7.33 (m, 3H), 7.30–7.25 (m, 2H), 6.03–5.93 (m, 2H), 5.18 (t, *J* = 7.2 Hz, 1H), 4.51 (d, *J* = 5.6 Hz, 2H), 3.17 (t, *J* = 7.5 Hz, 2H), 2.21–2.12 (m, 2H),1.64 (s, 3H), 1.52 (s, 3H), 1.34 (s, 9H). ^13^C NMR (75 MHz, CDCl_3_) δ: 166.00, 154.15, 150.21, 140.82, 140.16, 140.10, 135.08, 131.94, 128.50, 127.47, 127.18, 126.76, 126.66, 125.31, 123.47, 120.13, 42.96, 34.19, 31.03, 30.40, 27.18, 25.30, 17.38. HR-MS (ESI) *m/z*: Calcd. for C_32_H_37_NO [M+H]^+^, 452.2948; Found, 452.2950.

(2*E*)-7-Methyl-1-morpholino-3-(naphthalen-1-yl)octa-2,6-dien-1-one **10e**. A colorless liquid, 265 mg, yield 62%. ^1^H NMR (300 MHz, CDCl_3_) δ: 7.99 (dd, *J* = 6.3, 3.5 Hz, 1H), 7.86 (dd, *J* = 6.3, 3.5 Hz, 1H), 7.80 (d, *J* = 8.1 Hz, 1H), 7.52–7.41 (m, 3H), 7.30 (dd, *J* = 6.9, 1.2 Hz, 1H), 6.06 (s, 1H), 5.07 (t, *J* = 7.2 Hz, 1H), 3.79–3.53 (m, 8H), 2.87 (t, *J* = 7.8 Hz, 2H), 2.10–2.04 (m, 2H), 1.60 (s, 3H), 1.40 (s, 3H). ^13^C NMR (75 MHz, CDCl_3_) δ: 166.09, 151.85, 139.78, 133.40, 131.80, 130.80, 128.05, 127.51, 125.76, 125.52, 125.17, 124.72, 124.63, 123.34, 122.34, 66.63, 66.50, 46.42, 41.45, 34.14, 26.47, 25.29, 17.30. HR-MS (ESI) *m/z*: Calcd. for C_23_H_27_NO_2_ [M+H]^+^, 350.2115; Found, 350.2118.

(2*E*)-*N*-(4-*tert*-Butylbenzyl)-7-methyl-3-(naphthalen-1-yl)octa-2,6-dienamide **10f**. A colorless liquid, 338 mg, yield 65%. ^1^H NMR (300 MHz, CDCl_3_) δ: 7.96 (dd, *J* = 6.3, 3.5 Hz, 1H), 7.86 (dd, *J* = 6.3, 3.5 Hz, 1H), 7.80 (d, *J* = 8.2 Hz, 1H), 7.50–7.35 (m, 5H), 7.32–7.20 (m, 3H), 5.85 (t, *J* = 5.2 Hz, 1H), 5.81 (s, 1H), 5.11 (t, *J* = 7.2 Hz, 1H), 4.53 (d, *J* = 5.7 Hz, 2H), 3.20 (t, *J* = 7.8 Hz, 2H), 2.17–2.03 (m, 2H), 1.59 (s, 3H), 1.42 (s, 3H), 1.33 (s, 9H). ^13^C NMR (75 MHz, CDCl_3_) δ: 165.65, 155.61, 150.23, 140.41, 135.01, 133.37, 131.78, 130.57, 127.99, 127.45, 125.69, 125.49, 125.32, 124.69, 124.46, 123.52, 122.95, 42.91, 34.19, 33.44, 31.01, 26.65, 25.25, 17.28. HR-MS (ESI) *m/z*: Calcd. for C_30_H_35_NO [M+H]^+^, 426.2791; Found, 426.2788.

(2*E*)-7-Methyl-1-morpholino-3-(naphthalen-2-yl)octa-2,6-dien-1-one **10g**. A colorless liquid, 220 mg, yield 50%. ^1^H NMR (300 MHz, CDCl_3_) δ: 7.90–7.79 (m, 4H), 7.57–7.45 (m, 3H), 6.26 (s, 1H), 5.14 (t, *J* = 7.2 Hz, 1H), 3.80–3.53 (m, 8H), 2.88 (t, *J* = 8.1 Hz, 2H), 2.17–2.08 (m, 2H), 1.65 (s, 3H), 1.47 (s, 3H). ^13^C NMR (75 MHz, CDCl_3_) δ: 166.61, 150.71, 137.83, 132.95, 132.81, 131.90, 127.87, 127.80, 127.26, 126.07, 125.96, 125.33, 124.21, 123.23, 119.99, 66.58, 46.48, 41.45, 31.30, 26.95, 25.34, 17.35. HR-MS (ESI) *m/z*: Calcd. for C_23_H_27_NO_2_ [M+H]^+^, 350.2115; Found, 350.2118.

(2*E*)-*N*-(4-*tert*-Butyl)benzyl-7-methyl-3-(naphthalen-2-yl)octa-2,6-dienamide **10h**. A colorless liquid, 235 mg, yield 44%. ^1^H NMR (300 MHz, CDCl_3_) δ: 7.90–7.78 (m, 4H), 7.54–7.46 (m, 3H), 7.39 (d, *J* = 8.1 Hz, 2H), 7.29 (d, *J* = 8.1 Hz, 2H), 6.09 (s, 1H), 6.03 (s, 1H), 5.19 (t, *J* = 7.2 Hz, 1H), 4.52 (d, *J* = 5.6 Hz, 2H), 3.25 (t, *J* = 7.8 Hz, 2H), 2.21–2.12 (m, 2H), 1.63 (s, 3H), 1.49 (s, 3H), 1.34 (s, 9H). ^13^C NMR (75 MHz, CDCl_3_) δ: 166.03, 154.50, 150.19, 138.65, 135.09, 132.94, 132.89, 131.94, 127.96, 127.75, 127.48, 127.26, 126.03, 126.00, 125.56, 125.31, 124.30, 123.47, 120.77, 42.97, 34.19, 31.03, 30.50, 27.19, 25.30, 17.37. HR-MS (ESI) *m/z*: Calcd. for C_30_H_35_NO [M+H]^+^, 426.2791; Found, 426.2788.

(2*E*)-*N*-(2-Chloro-4-trifluoromethylphenyl)-3-(4-(*tert*-butylphenyl)-7-methylocta-2,6-dienamide **10i**. A colorless liquid, 500 mg, yield 89%. ^1^H NMR (300 MHz, CDCl_3_) δ: 8.73 (d, *J* = 8.7 Hz, 1H), 7.83 (s, 1H), 7.65 (d, *J* = 1.5 Hz, 1H), 7.54 (dd, *J* = 8.7, 1.5 Hz, 1H), 7.43 (s, 4H), 6.11 (s, 1H), 5.19 (t, *J* = 7.2 Hz, 1H), 3.20 (t, *J* = 7.8 Hz, 2H), 2.25–2.16 (m, 2H), 1.65 (s, 3H), 1.55 (s, 3H), 1.36 (s, 9H). ^13^C NMR (75 MHz, CDCl_3_) δ: 163.99, 159.87, 152.09, 137.93, 137.85, 131.96, 126.07, 125.77 (q, *J* = 4.0 Hz), 125.21, 124.56 (q, *J* = 3.6 Hz), 123.26, 121.83, 121.26, 120.47, 118.40, 77.11, 76.68, 76.26, 34.35, 30.90, 30.70, 27.42, 25.30, 17.34. HR-MS (ESI) *m/z*: Calcd. for C_26_H_29_ClF_3_NO [M+H]^+^, 464.1963; Found, 464.1960.

(2*E*)-*N*-(2-(4-Chlorophenyl)phenyl)-3-(4-(*tert*-butyl)phenyl)-7-methylocta-2,6-dienamide **10j**. A colorless liquid, 340 mg, yield 61%. ^1^H NMR (300 MHz, CDCl_3_) δ: 8.39 (s, 1H), 7.35–7.20 (m, 12H), 5.78 (s, 1H), 5.16 (t, *J* = 7.2 Hz, 1H), 3.14 (t, *J* = 7.8 Hz, 2H), 2.18–2.09 (m, 2H), 1.63 (s, 3H), 1.52 (s, 3H), 1.32 (s, 9H). ^13^C NMR (75 MHz, CDCl_3_) δ: 163.87, 157.77, 151.62, 138.21, 136.38, 134.70, 133.75, 131.66, 130.30, 129.67, 128.99, 128.39, 126.00, 125.08, 123.99, 123.52, 118.75, 34.28, 30.90, 30.55, 27.40, 25.28, 17.35. HR-MS (ESI) *m/z*: Calcd. for C_31_H_34_ClNO [M+H]^+^, 472.2402; Found, 472.2406.

(2*E*)-*N*-(2-Chloro-4-trifluoromethylphenyl)-3-(4-phenylphenyl)-7-methylocta-2,6-dienamide **10k**. A colorless liquid, 309 mg, yield 53%. ^1^H NMR (300 MHz, CDCl_3_) δ: 8.75 (d, *J* = 8.7 Hz, 1H), 7.87 (s, 1H), 7.67–7.40 (m, 11H), 6.18 (s, 1H), 5.21 (t, *J* = 7.0 Hz, 1H), 3.25 (t, *J* = 7.8 Hz, 2H), 2.28–2.20 (m, 2H), 1.66 (s, 3H), 1.56 (s, 3H). ^13^C NMR (75 MHz, CDCl_3_) δ: 163.88, 159.46, 141.59, 139.91, 139.77, 137.78, 132.13, 128.57, 127.38, 126.95, 126.84, 126.70, 125.82 (q, *J* = 4.1 Hz), 124.60 (q, *J* =3.6 Hz), 123.13, 121.90, 120.53, 118.93, 30.71, 27.37, 25.32, 17.39. HR-MS (ESI) *m/z*: Calcd. for C_28_H_25_ClF_3_NO [M+H]^+^, 484.1650; Found, 484.1646.

(2*E*)-*N*-(2-(4-Chlorophenyl)phenyl)-3-(4-phenylphenyl)-7-methylocta-2,6-dienamide **10l**. A colorless liquid, 348 mg, yield 60%. ^1^H NMR (300 MHz, CDCl_3_) δ: 8.39 (s, 1H), 7.64–7.56 (m, 4H), 7.49–7.20 (m, 13H), 5.85 (s, 1H), 5.18 (t, *J* = 7.0 Hz, 1H), 3.19 (t, *J* = 7.8 Hz, 2H), 2.21–2.10 (m, 2H), 1.64 (s, 3H), 1.53 (s, 3H). ^13^C NMR (75 MHz, CDCl_3_) δ: 157.38, 141.20, 140.09, 140.01, 136.38, 134.61, 133.78, 131.82, 130.31, 129.72, 129.00, 128.50, 128.40, 127.24, 126.83, 126.77, 126.67, 124.13, 123.38, 119.49, 30.55, 27.34, 25.29, 17.37. HR-MS (ESI) *m/z*: Calcd. for C_33_H_30_ClNO [M+H]^+^, 492.2089; Found, 492.2085.

(2*E*)-*N*-(2-Chloro-4-trifluoromethylphenyl)-7-methyl-3-(naphthalen-1-yl)octa-2,6-dienamide **10m**. A colorless liquid, 365 mg, yield 66%. ^1^H NMR (300 MHz, CDCl_3_) δ: 8.79 (d, *J* = 8.7 Hz, 1H), 8.01–7.78 (m, 4H), 7.65 (s, 1H), 7.52 (m, 4H), 7.32 (d, *J* = 6.2 Hz, 1H), 6.04 (s, 1H), 5.13 (t, *J* = 7.0 Hz, 1H), 3.28 (t, *J* = 7.7 Hz, 2H), 2.24–2.16 (m, 2H), 1.61 (s, 3H), 1.45 (s, 3H). ^13^C NMR (75 MHz, CDCl_3_) δ: 163.60, 160.90, 140.04, 137.74, 133.44, 132.06, 130.37, 128.15, 127.98, 125.98, 125.82 (q, *J* = 4.0 Hz), 125.69, 125.42, 125.07, 124.70, 124.61 (q, *J* = 3.8 Hz), 124.40, 123.19, 122.16, 121.93, 121.25, 120.50, 76.27, 33.80, 26.71, 25.26, 17.29. HR-MS (ESI) *m/z*: Calcd. for C_26_H_23_ClF_3_NO [M-H]^-^, 456.1348; Found, 456.1366.

(2*E*)-*N*-(2-(4-Chlorophenyl)phenyl)-7-methyl-3-(naphthalen-1-yl)octa-2,6-dienamide **10n**. A colorless liquid, 430 mg, yield 70%. ^1^H NMR (300 MHz, CDCl_3_) δ: 8.46 (d, *J* = 7.1 Hz, 1H), 7.92–7.78 (m, 3H), 7.51–7.39 (m, 6H), 7.35–7.29 (m, 2H), 7.21 (m, 3H), 7.11 (s, 1H), 5.70 (s, 1H), 5.10 (t, *J* = 7.0 Hz, 1H), 3.21 (t, *J* = 7.8 Hz, 2H), 2.18–2.10 (m, 2H), 1.59 (s, 3H), 1.42 (s, 3H). ^13^C NMR (75 MHz, CDCl_3_) δ: 163.80, 158.34, 140.22, 136.27, 134.59, 133.80, 133.37, 131.77, 130.44, 130.26, 129.66, 128.99, 128.44, 128.06, 127.69, 125.79, 125.58, 125.11, 124.68, 124.39, 124.12, 123.42, 122.67, 121.44, 33.54, 26.70, 25.25, 17.29. HR-MS (ESI) *m/z*: Calcd. for C_31_H_28_ClNO [M+H]^+^, 466.1932; Found, 466.1928.

(2*E*)-*N*-(2-Chloro-4-trifluoromethylphenyl)-7-methyl-3-(naphthalene-2-yl)octa-2,6-dienamide **10o**. A colorless liquid, 294 mg, yield 53%. ^1^H NMR (300 MHz, CDCl_3_) δ: 8.75 (d, *J* = 8.8 Hz, 1H), 7.98–7.82 (m, 5H), 7.66 (s, 1H), 7.58–7.50 (m, 4H), 6.25 (s, 1H), 5.22 (t, *J* = 7.1 Hz, 1H), 3.27 (t, *J* = 7.8 Hz, 2H), 2.25–2.16 (m, 2H), 1.65 (s, 3H), 1.52 (s, 3H). ^13^C NMR (75 MHz, CDCl_3_) δ: 163.88, 159.83, 138.26, 137.78, 133.17, 132.88, 132.13, 128.07, 128.00, 127.31, 126.40, 126.26, 125.88, 125.82 (q, *J* = 4.1 Hz), 125.20, 124.60 (q, *J* = 3.5 Hz), 124.02, 123.13, 121.92, 120.55, 119.52, 30.83, 27.37, 25.30, 17.36. HR-MS (ESI) *m/z*: Calcd. for C_26_H_23_ClF_3_NO [M+H]^+^, 458.1493; Found, 458.1488.

(2*E*)-*N*-(2-(4-Chlorophenyl)phenyl)-7-methyl-3-(naphthalen-2-yl)octa-2,6-dienamide **10p**. A colorless liquid, 424 mg, yield 69%. ^1^H NMR (300 MHz, CDCl_3_) δ: 8.40 (s, 1H), 7.84 (brs, 4H), 7.52–7.20 (m, 11H), 5.93 (s, 1H), 5.19 (t, *J* = 6.9 Hz, 1H), 3.27 (t, *J* = 7.8 Hz, 2H), 2.22–2.13 (m, 2H), 1.64 (s, 3H), 1.50 (s, 3H). ^13^C NMR (75 MHz, CDCl_3_) δ: 164.06, 157.75, 138.58, 136.40, 134.60, 133.78, 132.99, 132.85, 131.83, 130.31, 129.75, 129.00, 128.41, 127.99, 127.84, 127.26, 126.18, 126.14, 125.68, 124.16, 123.40, 121.82, 120.11, 30.69, 27.35, 25.30, 17.37. HR-MS (ESI) *m/z*: Calcd. for C_31_H_28_ClNO [M+H]^+^, 466.1932; Found, 466.1928.

### 3.5. Synthesis of the Chiral Amides ***11a***–***11p***

The general synthetic method: A 50 mL round-bottomed flask equipped with a magnetic stirring bar was charged with AD-mix-β (1.4 g), water (7.5 mL) and *tert*-butyl alcohol (7.5 mL). The resulting mixture was stirred at room temperature to produce two clear phases. Methanesulfonamide (68 mg, 0.7 mmol) was added in one portion and the reaction mixture was stirred for 1.5 h. The reaction mixture was cooled to 0 °C. Compound **10a** (356 mg, 1.0 mmol) was added at once, and the heterogeneous slurry was stirred vigorously at 0 °C for 40 h. The saturated Na_2_S_2_O_3_ solution (15 mL) was added at 0 °C, and the mixture was allowed to reach room temperature and stirred for 30 min. EtOAc (50 mL) and water (20 mL) were added to the reaction mixture. The organic layer was separated and the aqueous layer was re-extracted with EtOAc (50 mL×3). The combined organic phase was dried over with anhydrous Na_2_SO_4_ and the solvent was removed to give the crude product. This product was purified by flash chromatography on silica gel with petroleum ether/EtOAc (V:V = 3:1) as the eluent to give a colorless oil chiral diol amide (*R*)-**11a** 335 mg, yield 86%. In a similar way, the chiral diol amides (*R*)-**11b**-(*S*)-**11p** were prepared.

(6*R*,2*E*)-3-(4-(*tert*-Butylphenyl)-6,7-dihydroxy-7-methyl-1-morpholinooct-2-en-1-one (*R*)-**11a**. A colorless liquid, 335 mg, yield 86%. [α${]}_{D}^{20}$ = − 77.8 (c 1.0, CHCl_3_), ee 95.4%. ^1^H NMR (300 MHz, CDCl_3_) δ: 7.39 (d, *J* = 8.4 Hz, 2H), 7.31 (d, *J* = 8.4 Hz, 2H), 6.19 (s, 1H), 3.88–3.44 (m, 9H), 3.33–3.27 (m, 1H), 3.23–3.10 (m, 1H), 2.75–2.69 (m, 1H), 1.50–1.40 (m, 2H), 1.32 (s, 9H), 1.01 (s, 3H), 0.92 (s, 3H). ^13^C NMR (75 MHz, CDCl_3_) δ: 166.84, 152.89, 151.69, 136.66, 126.04, 125.24, 119.38, 74.67, 71.64, 66.52, 66.37, 46.44, 41.72, 34.30, 30.90, 28.45, 26.98, 25.55, 23.01. HR-MS (ESI) *m/z*: Calcd. for C_23_H_35_NO_4_ [M+H]^+^, 390.2639; Found, 390.2636.

(6*S*,2*E*)-3-(4-*tert*-Butylphenyl)-6,7-dihydroxy-7-methyl-1-morpholinooct-2-en-1-one (***S***)-**11a**. A colorless liquid, 120 mg, yield 89%. [α${]}_{D}^{20}$ = +77.0 (c 0.9, CHCl_3_), ee 91.0%. ^1^H NMR (300 MHz, CDCl_3_) δ: 7.39 (d, *J* = 8.4 Hz, 2H), 7.31 (d, *J* = 8.4 Hz, 2H), 6.19 (s, 1H), 3.88–3.44 (m, 9H), 3.33–3.27 (m, 1H), 3.23–3.10 (m, 1H), 2.75–2.69 (m, 1H), 1.50–1.40 (m, 2H), 1.32 (s, 9H), 1.01 (s, 3H), 0.92 (s, 3H). ^13^C NMR (75 MHz, CDCl_3_) δ: 166.84, 152.89, 151.69, 136.66, 126.04, 125.24, 119.38, 74.66, 71.64, 66.52, 66.37, 46.44, 41.72, 34.30, 30.90, 28.45, 26.98, 25.55, 23.01. HR-MS (ESI) *m/z*: Calcd. for C_23_H_35_NO_4_ [M+H]^+^, 390.2639; Found, 390.2636.

(6*R*,2*E*)-*N*-(4-*tert*-Butylbenzyl)-3-(4-*tert*-butylphenyl)-6,7-dihydroxy-7-methyloct-2-enamide (*R*)-**11b**. A colorless liquid, 105 mg, yield 78%. [α${]}_{D}^{20}$ = − 93.6 (c 2.6, CHCl_3_), ee 97.2%. ^1^H NMR (300 MHz, CDCl_3_) δ: 7.40–7.24 (m, 8H), 6.40 (brs, 1H), 6.02 (s, 1H), 5.66 (brs, 1H), 4.47 (d, *J* = 5.7 Hz, 2H), 3.71–3.56 (m, 1H), 3.34 (brs, 1H), 2.98 (brs, 1H), 2.82–2.76 (m, 1H), 1.55–1.47 (m, 2H), 1.33 (s, 18H), 1.04 (s, 3H), 0.99 (s, 3H). ^13^C NMR (75 MHz, CDCl_3_) δ: 166.89, 155.20, 151.69, 150.24, 137.41, 134.68, 127.45, 126.06, 125.31, 125.21, 120.29, 75.06, 71.86, 43.05, 34.29, 34.19, 31.03, 30.93, 29.20, 26.43, 25.70, 23.12. HR-MS (ESI) *m/z*: Calcd. for C_30_H_43_NO_3_ [M+H]^+^, 466.3316; Found, 466.3315.

(6*S*,2*E*)-*N*-(4-*tert*-Butylbenzyl)-3-(4-*tert*-butylphenyl)-6,7-dihydroxy-7-methyloct-2-enamide (***S***)-**11b**. A colorless solid, 116 mg, yield 86%, m.p. 136–138 °C. [α${]}_{D}^{20}$ = +94.0 (c 1.8, CHCl_3_), ee 91.4%. ^1^H NMR (300 MHz, CDCl_3_) δ: 7.40–7.24 (m, 8H), 6.09 (brs, 1H), 5.99 (s, 1H), 5.65 (d, *J* = 3.0 Hz, 1H), 4.48 (d, *J* = 5.7 Hz, 2H), 3.74–3.57 (m, 1H), 3.34 (d, *J* = 9.6 Hz, 1H), 2.94 (brs, 1H), 2.82–2.76 (m, 1H), 1.55–1.48 (m, 2H), 1.33 (s, 9H), 1.32 (s, 9H), 1.06 (s, 3H), 0.99 (s, 3H). ^13^C NMR (75 MHz, CDCl_3_) δ: 166.80, 155.52, 151.76, 150.37, 137.38, 134.57, 127.46, 126.03, 125.36, 125.23, 120.16, 74.97, 71.80, 43.11, 34.30, 34.20, 31.00, 30.91, 29.17, 26.45, 25.67, 23.06. HR-MS (ESI) *m/z*: Calcd. for C_30_H_43_NO_3_ [M+H]^+^, 466.3316; Found, 466.3315.

(6*R*,2*E*)-3-(4-Phenylphenyl)-6,7-dihydroxy-7-methyl-1-morpholinooct-2-en-1-one (*R*)-**11c**. A colorless solid, 115 mg, yield 84%, m.p. 135–136 °C. [α${]}_{D}^{20}$ = − 54.4 (c 1.7, CHCl_3_), ee 95.0%. ^1^H NMR (300 MHz, CDCl_3_) δ: 7.65–7.58 (m, 4H), 7.49–7.43 (m, 4H), 7.37 (t, *J* = 7.2 Hz, 1H), 6.26 (s, 1H), 5.94 (d, *J* = 3.6 Hz, 1H), 3.85–3.50 (m, 8H), 3.35–3.15 (m, 2H), 2.93 (s, 1H), 2.80–2.70 (m, 1H), 1.55–1.49 (m, 2H), 1.03 (s, 3H), 0.94 (s, 3H).^13^C NMR (75 MHz, CDCl_3_) δ: 166.71, 152.45, 141.25, 139.84, 138.56, 128.54, 127.34, 127.00, 126.82, 126.65, 120.00, 74.72, 71.66, 66.52, 66.39, 46.48, 41.75, 28.46, 27.05, 25.64, 23.03. HR-MS (ESI) *m/z*: Calcd. for C_25_H_31_NO_4_ [M+H]^+^, 410.2326; Found, 410.2325.

(6*S*,2*E*)-3-(4-Phenylphenyl)-6,7-dihydroxy-7-methyl-1-morpholinooct-2-en-1-one (***S***)-**11c**. A colorless solid, 116 mg, yield 85%, m.p. 134–135 °C. [α${]}_{D}^{20}$ = +54.3 (c 1.3, CHCl_3_), ee 90.4%. ^1^H NMR (300 MHz, CDCl_3_) δ: 7.66–7.58 (m, 4H), 7.50–7.43 (m, 4H), 7.37 (t, *J* = 7.2 Hz, 1H), 6.26 (s, 1H), 5.94 (d, *J* = 3.6 Hz, 1H), 3.88–3.51 (m, 8H), 3.35–3.15 (m, 2H), 2.93 (s, 1H), 2.80–2.70 (m, 1H), 1.55–1.49 (m, 2H), 1.03 (s, 3H), 0.94 (s, 3H). ^13^C NMR (75 MHz, CDCl_3_) δ: 166.71, 152.46, 141.25, 139.84, 138.56, 128.54, 127.34, 127.00, 126.82, 126.65, 120.00, 74.72, 71.67, 66.52, 66.39, 46.48, 41.75, 28.45, 27.04, 25.63, 23.02. HR-MS (ESI) *m/z*: Calcd. for C_25_H_31_NO_4_ [M+H]^+^, 410.2326; Found, 410.2325.

(6*R*,2*E*)-*N*-(4-*tert*-Butylbenzyl)-3-(4-phenylphenyl)-6,7-dihydroxy-7-methyloct-2-enamide (***R***)-**11d**. A colorless solid, 147 mg, yield 98%, m.p. 71–73 °C. [α${]}_{D}^{20}$ = −73.5 (c 1.8, CHCl_3_), ee 98.4%. ^1^H NMR (300 MHz, CDCl_3_) δ: 7.63–7.58 (m, 4H), 7.49–7.36 (m, 7H), 7.30–7.24 (m, 2H), 6.14 (t, *J* = 5.4 Hz, 1H), 6.06 (s, 1H), 5.67 (d, *J* = 3.0 Hz, 1H), 4.50 (d, *J* = 5.4 Hz, 2H), 3.84–3.62 (m, 1H), 3.37 (d, *J* = 10.1 Hz, 1H), 2.94 (s, 1H), 2.85–2.78 (m, 1H), 1.63–1.45 (m, 2H), 1.33 (s, 9H), 1.07 (s, 3H), 1.00 (s, 3H). ^13^C NMR (75 MHz, CDCl_3_) δ: 166.65, 155.05, 150.43, 141.30, 139.83, 139.28, 134.49, 128.54, 127.50, 127.33, 126.95, 126.80, 126.65, 125.38, 120.80, 74.98, 71.82, 43.17, 34.21, 31.00, 29.09, 26.47, 25.72, 23.04. HR-MS (ESI) *m/z*: Calcd. for C_32_H_39_NO_3_ [M+H]^+^, 486.3003; Found, 486.298.

(6*S*,2*E*)-N-(4-*tert*-Butylbenzyl)-3-(4-phenylphenyl)-6,7-dihydroxy-7-methyloct-2-enamide (***S***)-**11d**. A colorless solid, 148 mg, yield 98%, m.p. 148–149 °C. [α${]}_{D}^{20}$ = + 75.2 (c 0.8, CHCl_3_), ee 96.0%. ^1^H NMR (300 MHz, CDCl_3_) δ: 7.63–7.58 (m, 4H), 7.49–7.36 (m, 7H), 7.30–7.24 (m, 2H), 6.14 (t, *J* = 5.4 Hz, 1H), 6.06 (s, 1H), 5.67 (d, *J* = 3.0 Hz, 1H), 4.50 (d, *J* = 5.4 Hz, 2H), 3.84–3.62 (m, 1H), 3.37 (d, *J* = 10.1 Hz, 1H), 2.94 (s, 1H), 2.85–2.78 (m, 1H), 1.63–1.45 (m, 2H), 1.33 (s, 9H), 1.07 (s, 3H), 1.00 (s, 3H). ^13^C NMR (75 MHz, CDCl_3_) δ: 166.61, 155.19, 150.47, 141.33, 139.83, 139.26, 134.45, 128.53, 127.49, 127.33, 126.96, 126.79, 126.65, 125.40, 120.74, 74.95, 71.79, 43.19, 34.21, 30.99, 29.07, 26.48, 25.71, 23.02. HR-MS (ESI) *m/z*: Calcd. for C_32_H_39_NO_3_ [M+H]^+^, 486.3003; Found, 486.298. 

(6*R*,2*E*)-3-(Naphthalen-1-yl)-6,7-dihydroxy-7-methyl-1-morpholinooct-2-en-1-one (***R***)-**11e**. A colorless liquid, 101 mg, yield 91%. [α${]}_{D}^{20}$ = −67.7 (c 1.4, CHCl_3_), ee 97.6%. ^1^H NMR (300 MHz, CDCl_3_) δ: 8.03–7.94 (m, 1H), 7.92–7.78 (m, 2H), 7.55–7.42 (m, 3H), 7.32 (d, *J* = 6.9 Hz, 1H), 6.21 (s, 1H), 5.99 (d, *J* = 3.7 Hz, 1H), 3.85–3.59 (m, 7H), 3.52 (t, *J* = 4.7 Hz, 2H), 3.29 (td, *J* = 13.2, 4.3 Hz, 1H), 3.02 (s, 1H), 2.76 (dt, *J* = 13.4, 4.6 Hz, 1H), 1.86 (s, 1H), 1.50–1.27 (m, 2H), 1.02 (s, 3H), 1.02 (s, 3H). ^13^C NMR (75 MHz, CDCl_3_) δ: 166.38, 153.20, 138.67, 133.53, 130.47, 128.31, 128.00, 126.05, 125.71, 124.79, 124.70, 124.68, 122.84, 75.09, 71.78, 66.51, 66.32, 46.46, 41.77, 30.15, 28.48, 25.71, 22.99. HR-MS (ESI) *m/z*: Calcd. for C_23_H_29_NO_4_ [M+H]^+^, 384.2169; Found, 384.2168.

(6*S*,2*E*)-3-(Naphthalen-1-yl)-6,7-dihydroxy-7-methyl-1-morpholinooct-2-en-1-one (***S***)-**11e**. A colorless liquid, 102 mg, yield 91%. [α${]}_{D}^{20}$=+67.9 (c 1.3, CHCl_3_), ee 98.0%. ^1^H NMR (300 MHz, CDCl_3_) δ: 8.04–7.94 (m, 1H), 7.91–7.80 (m, 2H), 7.54–7.45 (m, 3H), 7.32 (d, *J* = 6.9 Hz, 1H), 6.21 (s, 1H), 5.99 (d, *J* = 3.7 Hz, 1H), 3.86–3.59 (m, 7H), 3.52 (t, *J* = 4.8 Hz, 2H), 3.36–3.22 (m, 1H), 3.03 (s, 1H), 2.76 (dt, *J* = 13.2, 4.5 Hz, 1H), 1.86 (s, 1H), 1.51–1.27 (m, 2H), 1.02 (s, 3H), 1.02 (s, 3H). ^13^C NMR (75 MHz, CDCl_3_) δ: 166.38, 153.18, 138.66, 133.53, 130.47, 128.31, 128.00, 126.06, 125.71, 124.79, 124.70, 124.68, 122.83, 75.11, 71.82, 66.52, 66.32, 46.46, 41.78, 30.15, 28.49, 25.73, 22.99. HR-MS (ESI) *m/z*: Calcd. for C_23_H_29_NO_4_ [M+H]^+^, 384.2169; Found, 384.2168.

(6*R*,2*E*)-*N*-(4-*tert*-Butylbenzyl)-3-(naphthalen-1-yl)-6,7-dihydroxy-7-methyloct-2-enamide (***R***)-**11f**. A colorless liquid, 123 mg, yield 95%. [α${]}_{D}^{20}$ = −76.2 (c 2.2, CHCl_3_), ee 95.6%. ^1^H NMR (300 MHz, CDCl_3_) δ: 7.95–7.77 (m, 3H), 7.537.36 (m, 5H), 7.28–7.20 (m, 3H), 6.14 (brs, 1H), 5.92 (s, 1H), 5.74 (d, *J* = 3.4 Hz, 1H), 4.50 (d, *J* = 5.6 Hz, 2H), 3.80–3.75 (m, 1H), 3.60 (d, *J* = 9.6 Hz, 1H), 3.03 (s, 1H), 2.80–2.70 (m, 1H), 1.50–1.35 (m, 2H), 1.33 (s, 9H), 1.05 (s, 3H), 1.04 (s, 3H). ^13^C NMR (75 MHz, CDCl_3_) δ: 166.39, 155.67, 150.45, 139.56, 134.41, 133.46, 130.34, 128.19, 127.91, 127.52, 125.98, 125.72, 125.39, 124.94, 124.64, 124.55, 123.80, 75.00, 71.92, 43.15, 34.21, 31.01, 29.55, 28.84, 25.79, 22.99. HR-MS (ESI) *m/z*: Calcd. for C_30_H_37_NO_3_ [M+H]^+^, 460.2846; Found, 460.2840.

(6*S*,2*E*)-*N*-(4-*tert*-Butylbenzyl)-3-(naphthalen-1-yl)-6,7-dihydroxy-7-methyloct-2-enamide (***S***)-**11f**. A colorless liquid, 120 mg, yield 93%. [α${]}_{D}^{20}$ = +77.5 (c 2.1, CHCl_3_), ee 97.0%. ^1^H NMR (300 MHz, CDCl_3_) δ: 7.95–7.79 (m, 3H), 7.53–7.36 (m, 5H), 7.28–7.20 (m, 3H), 6.10 (brs, 1H), 5.93 (s, 1H), 5.73 (d, *J* = 3.4 Hz, 1H), 4.50 (d, *J* = 5.6 Hz, 2H), 3.80–3.75 (m, 1H), 3.60 (d, *J* = 9.6 Hz, 1H), 3.03 (s, 1H), 2.80–2.70 (m, 1H), 1.50–1.36 (m, 2H), 1.33 (s, 9H), 1.06 (s, 3H), 1.05 (s, 3H). ^13^C NMR (75 MHz, CDCl_3_) δ: 166.38, 155.68, 150.45, 139.56, 134.40, 133.46, 130.34, 128.19, 127.91, 127.52, 125.98, 125.72, 125.40, 124.94, 124.64, 124.55, 123.79, 74.99, 71.91, 43.15, 34.21, 31.01, 29.55, 28.84, 25.79, 22.99. HR-MS (ESI) *m/z*: Calcd. for C_30_H_37_NO_3_ [M+H]^+^, 460.2846; Found, 460.2840.

(6*R*,2*E*)-3-(Naphthalen-2-yl)-6,7-dihydroxy-7-methyl-1-morpholinooct-2-en-1-one (***R***)-**11g**. A colorless liquid, 102 mg, yield 93%. [α${]}_{D}^{20}$ = −54.7 (c 2.5, CHCl_3_), ee 95.4%. ^1^H NMR (300 MHz, CDCl_3_) δ: 7.87–7.82 (m, 4H), 7.52–7.48 (m, 3H), 6.32 (s, 1H), 5.94 (d, *J* = 3.4 Hz, 1H), 3.85–3.50 (m, 8H), 3.44–3.26 (m, 2H), 2.97 (s, 1H), 2.86 (dt, *J* = 13.8, 4.3 Hz, 1H), 1.59–1.43 (m, 2H), 0.99 (s, 3H), 0.87 (s, 3H). ^13^C NMR (75 MHz, CDCl_3_) δ: 166.72, 152.81, 137.08, 133.00, 132.89, 128.16, 127.88, 127.33, 126.27, 126.25, 125.61, 124.10, 120.64, 74.79, 71.67, 66.51, 66.39, 46.48, 41.76, 28.51, 27.17, 25.60, 22.98. HR-MS (ESI) *m/z*: Calcd. for C_23_H_29_NO_4_ [M+H]^+^, 384.2169; Found, 384.2163.

(6*S*,2*E*)-3-(Naphthalen-2-yl)-6,7-dihydroxy-7-methyl-1-morpholinooct-2-en-1-one (***S***)-**11g**. A colorless liquid, 101 mg, yield 92%. [α${]}_{D}^{20}$ = +54.1 (c 2.5, CHCl_3_), ee 92.6%. ^1^H NMR (300 MHz, CDCl_3_) δ: 7.88–7.82 (m, 4H), 7.52–7.48 (m, 3H), 6.32 (s, 1H), 5.94 (d, *J* = 3.4 Hz, 1H), 3.86–3.49 (m, 8H), 3.44–3.26 (m, 2H), 2.97 (s, 1H), 2.86 (dt, *J* = 13.6, 4.2 Hz, 1H), 1.58–1.44 (m, 2H), 0.99 (s, 3H), 0.87 (s, 3H). ^13^C NMR (75 MHz, CDCl_3_) δ: 166.71, 152.79, 137.10, 133.00, 132.89, 128.15, 127.88, 127.33, 126.27, 126.25, 125.61, 124.10, 120.63, 74.81, 71.66, 66.51, 66.38, 46.48, 41.76, 28.53, 27.18, 25.60, 23.01. HR-MS (ESI) *m/z*: Calcd. for C_23_H_29_NO_4_ [M+H]^+^, 384.2169; Found, 384.2163.

(6*R*,2*E*)-*N*-(4-*tert*-Butylbenzyl)-3-(naphthalen-2-yl)-6,7-dihydroxy-7-methyloct-2-enamide (***R***)-**11h**. A colorless liquid, 97 mg, yield 87%. [α${]}_{D}^{20}$= −76.1 (c 2.1, CHCl_3_), ee 96.4%. ^1^H NMR (300 MHz, CDCl_3_) δ: 7.86–7.80 (m, 4H), 7.56–7.43 (m, 3H), 7.39 (d, *J* = 8.2 Hz, 2H), 7.27 (d, *J* = 8.2 Hz, 2H), 6.33 (brs, 1H), 6.13 (s, 1H), 5.69 (s, 1H), 4.50 (d, *J* = 5.7 Hz, 2H), 3.80–3.71 (m, 1H), 3.40 (d, *J* = 10.5 Hz, 1H), 2.97–2.87 (m, 2H), 1.61–1.42 (m, 2H), 1.33 (s, 9H), 1.03 (s, 3H), 0.94 (s, 3H). ^13^C NMR (75 MHz, CDCl_3_) δ: 166.62, 155.56, 150.48, 137.79, 134.44, 134.44, 133.08, 132.89, 128.10, 127.95, 127.50, 127.30, 126.28, 126.23, 125.72, 125.40, 124.07, 121.41, 74.98, 71.78, 43.2, 34.21, 30.99, 29.06, 26.58, 25.68, 22.97. HR-MS (ESI) *m/z*: Calcd. for C_30_H_37_NO_3_ [M+H]^+^, 460.2846; Found, 460.2845.

(6*S*,2*E*)-*N*-(4-*tert*-Butylbenzyl)-3-(naphthalen-2-yl)-6,7-dihydroxy-7-methyloct-2-enamide (***S***)-**11h**. A colorless liquid, 102 mg, yield 92%. [α${]}_{D}^{20}$ = +77.5 (c 2.5, CHCl_3_), ee 90.6%. ^1^H NMR (300 MHz, CDCl_3_) δ: 7.86–7.80 (m, 4H), 7.55–7.43 (m, 3H), 7.39 (d, *J* = 8.2 Hz, 2H), 7.27 (d, *J* = 8.2 Hz, 2H), 6.33 (brs, 1H), 6.13 (s, 1H), 5.69 (s, 1H), 4.50 (d, *J* = 5.7 Hz, 2H), 3.80–3.70 (m, 1H), 3.40 (d, *J* = 10.5 Hz, 1H), 2.97–2.87 (m, 2H), 1.60–1.42 (m, 2H), 1.33 (s, 9H), 1.03 (s, 3H), 0.94 (s, 3H). ^13^C NMR (75 MHz, CDCl_3_) δ: 166.67, 155.35, 150.41, 137.81, 134.50, 133.07, 132.89, 128.10, 127.97, 127.50, 127.31, 126.28, 126.24, 125.72, 125.38, 124.10, 124.10, 121.50, 75.01, 71.82, 43.17, 34.21, 31.01, 29.07, 26.56, 25.69, 22.99. HR-MS (ESI) *m/z*: Calcd. for C_30_H_37_NO_3_ [M+H]^+^, 460.2846; Found, 460.2845.

(6*R*,2*E*)-*N*-(2-Chloro-4-trifluoromethylphenyl)-3-(4-*tert*-butylphenyl)-6,7-dihydroxy-7-methyloct-2-enamide (***R***)-**11i**. A colorless liquid, 100 mg, yield 78%. [α${]}_{D}^{20}$ = −111.7 (c 1.5, CHCl_3_), ee 97.6%. ^1^H NMR (300 MHz, CDCl_3_) δ: 8.66 (d, *J* = 8.7 Hz, 1H), 7.93 (s, 1H), 7.65 (s, 1H), 7.54 (d, *J* = 8.7 Hz, 1H), 7.43 (d, *J* = 8.7 Hz, 2H), 7.39 (d, *J* = 8.7 Hz, 2H), 6.20 (s, 1H), 4.81 (d, *J* = 3.2 Hz, 1H), 3.64–3.56 (m, 1H), 3.41–3.35 (m, 1H), 2.99–2.90 (m, 1H), 2.74 (s, 1H), 1.63–1.52 (m, 2H), 1.34 (s, 9H), 1.08 (s, 3H), 1.03 (s, 3H). ^13^C NMR (75 MHz, CDCl_3_) δ: 165.07, 160.06, 152.65, 137.25, 136.97, 126.10, 125.87 (q, *J* =3.9 Hz), 125.47, 124.62 (q, *J* = 3.7 Hz), 122.26, 120.92, 119.53, 75.59, 71.90, 34.40, 30.87, 29.57, 26.83, 25.65, 23.13. HR-MS (ESI) *m/z*: Calcd. for C_26_H_31_ClF_3_NO_3_ [M+H]^+^, 498.2017; Found, 498.2014.

(6*S*,2*E*)-*N*-(2-Chloro-4-trifluoromethylphenyl)-3-(4-*tert*-butylphenyl)-6,7-dihydroxy-7-methyloct-2-enamide (***S***)-**11i**. A colorless liquid, 99 mg, yield 77%. [α${]}_{D}^{20}$ = +111.5 (c 1.1, CHCl_3_), ee 97.2%. ^1^H NMR (300 MHz, CDCl_3_) δ: 8.67 (d, *J* = 8.7 Hz, 1H), 7.92 (s, 1H), 7.66 (s, 1H), 7.55 (d, *J* = 8.7 Hz, 1H), 7.44 (d, *J* = 8.7 Hz, 2H), 7.39 (d, *J* = 8.7 Hz, 2H), 6.20 (s, 1H), 4.81 (d, *J* = 3.5 Hz, 1H), 3.64–3.55 (m, 1H), 3.41–3.35 (m, 1H), 2.99–2.90 (m, 1H), 2.73 (s, 1H), 1.63–1.52 (m, 2H), 1.34 (s, 9H), 1.08 (s, 3H), 1.03 (s, 3H). ^13^C NMR (75 MHz, CDCl_3_) δ: 165.07, 160.07, 152.66, 137.24, 136.97, 126.10, 125.87 (q, *J* = 3.9 Hz), 125.47, 124.62 (q, *J* = 3.8 Hz), 122.24, 120.90, 119.54, 75.58, 71.89, 34.40, 30.87, 29.56, 26.83, 25.64, 23.12. HR-MS (ESI) *m/z*: Calcd. for C_26_H_31_ClF_3_NO_3_ [M+H]^+^, 498.2017; Found, 498.2014.

(6*R*,2*E*)-*N*-(2-(4-Chlorophenyl)phenyl)-3-(4-*tert*-butylphenyl)-6,7-dihydroxy-7-methyloct-2-enamide (***R***)-**11j**. A colorless liquid, 102 mg, yield 79%. [α${]}_{D}^{20}$ = −108.3 (c 0.9, CHCl_3_), ee 93.8%. ^1^H NMR (300 MHz, CDCl_3_) δ: 8.39 (d, *J* = 8.1 Hz, 1H), 7.46–7.21 (m, 12H), 5.85 (s, 1H), 5.26 (s, 1H), 3.65–3.58 (m, 1H), 3.34 (d, *J* = 9.0 Hz, 1H), 2.85–2.75 (m, 2H), 1.56–1.48 (m, 2H), 1.32 (s, 9H), 1.06 (s, 3H), 0.99 (s, 3H). ^13^C NMR (75 MHz, CDCl_3_) δ: 165.01, 157.78, 152.19, 137.20, 136.12, 134.12, 133.91, 130.97, 130.28, 129.73, 129.07, 128.46, 126.07, 125.34, 124.47, 121.62, 120.32, 75.17, 71.78, 34.34, 30.88, 29.24, 26.56, 25.63, 23.04. HR-MS (ESI) *m/z*: Calcd. for C_31_H_36_ClNO_3_ [M+H]^+^, 506.2456; Found, 506.2455.

(6*S*,2*E*)-*N*-(2-(4-Chlorophenyl)phenyl)-3-(4-*tert*-butylphenyl)-6,7-dihydroxy-7-methyloct-2-enamide (***S***)-**11j**. A colorless liquid, 105 mg, yield 82%. [α${]}_{D}^{20}$ = +106.1 (c 2.1, CHCl_3_), ee 90.5%. ^1^H NMR (300 MHz, CDCl_3_) δ: 8.38 (d, *J* = 8.1 Hz, 1H), 7.44–7.21 (m, 12H), 5.86 (s, 1H), 5.24 (s, 1H), 3.65–3.58 (m, 1H), 3.34 (d, *J* = 9.0 Hz, 1H), 2.85–2.75 (m, 2H), 1.55–1.48 (m, 2H), 1.32 (s, 9H), 1.06 (s, 3H), 0.99 (s, 3H). ^13^C NMR (75 MHz, CDCl_3_) δ: 165.05, 157.72, 152.17, 137.21, 136.16, 134.13, 133.89, 131.07, 130.28, 129.75, 129.06, 128.44, 126.09, 125.34, 124.51, 121.75, 120.32, 75.20, 71.80, 34.34, 30.89, 29.25, 26.57, 25.64, 23.07. HR-MS (ESI) *m/z*: Calcd. for C_31_H_36_ClNO_3_ [M+H]^+^, 506.2456; Found, 506.2455.

(6*R*,2*E*)-*N*-(2-Chloro-4-trifluoromethylphenyl)-3-(4-phenylphenyl)-6,7-dihydroxy-7-methyloct-2-enamide (***R***)-**11k**. A colorless liquid, 120 mg, yield 93%. [α${]}_{D}^{20}$ = −107.5 (c 1.5, CHCl_3_), ee 96.2%. ^1^H NMR (300 MHz, CDCl_3_) δ: 8.67 (d, *J* = 8.7 Hz, 1H), 7.98 (s, 1H), 7.68–7.35 (m, 11H), 6.27 (s, 1H), 4.81 (s, 1H), 3.75–3.58 (m, 1H), 3.45–3.33 (m, 1H), 3.05–2.94 (m, 1H), 2.74 (brs, 1H), 1.66–1.55 (m, 2H), 1.09 (s, 3H), 1.04 (s, 3H). ^13^C NMR (75 MHz, CDCl_3_) δ: 164.98, 159.54, 142.07, 139.67, 138.76, 137.19, 128.60, 127.52, 127.17, 126.87, 126.69, 125.91 (q, *J* = 3.6 Hz), 124.64 (q, 4.1 Hz), 122.36, 121.02, 120.05, 75.60, 71.92, 29.51, 26.86, 25.70, 23.12. HR-MS (ESI) *m/z*: Calcd. for C_28_H_27_ClF_3_NO_3_ [M+H]^+^, 518.1704; Found, 518.1702.

(6*S*,2*E*)-*N*-(2-Chloro-4-trifluoromethylphenyl)-3-(4-phenylphenyl)-6,7-dihydroxy-7-methyloct-2-enamide (***S***)-**11k**. A colorless liquid, 122 mg, yield 95%. [α${]}_{D}^{20}$ = +106.8 (c 1.1, CHCl_3_), ee 93.5%. ^1^H NMR (300 MHz, CDCl_3_) δ: 8.68 (d, *J* = 8.6 Hz, 1H), 7.98 (s, 1H), 7.68–7.36 (m, 11H), 6.27 (s, 1H), 4.81 (s, 1H), 3.74–3.59 (m, 1H), 3.41 (brs, 1H), 3.05–2.94 (m, 1H), 2.73 (s, 1H), 1.64–1.58 (m, 2H), 1.09 (s, 3H), 1.04 (s, 3H). ^13^C NMR (75 MHz, CDCl_3_) δ: 164.98, 159.55, 142.07, 139.67, 138.76, 137.19, 128.60, 127.52, 127.17, 126.87, 126.69, 125.91 (q, *J* = 3.8 Hz), 124.64 (q, 3.9 Hz), 122.35, 121.01, 120.05, 75.59, 71.92, 29.51, 26.86, 25.70, 23.11. HR-MS (ESI) *m/z*: Calcd. for C_28_H_27_ClF_3_NO_3_ [M+H]^+^, 518.1704; Found, 518.1702.

(6*R*,2*E*)-*N*-(2-(4-Chlorophenyl)phenyl)-3-(4-phenylphenyl)-6,7-dihydroxy-7-methyloct-2-enamide (***R***)-**11l**. A colorless liquid, 107 mg, yield 83%. [α${]}_{D}^{20}$ = −99.9 (c 1.3, CHCl_3_), ee 96.5%. ^1^H NMR (300 MHz, CDCl_3_) δ: 8.37 (d, *J* = 8.1 Hz, 1H), 7.64–7.23 (m, 17H), 5.94 (s, 1H), 5.22 (s, 1H), 3.75–3.64 (m, 1H), 3.36 (d, *J* = 9.6 Hz, 1H), 2.89–2.80 (m, 2H), 1.62–1.50 (m, 2H), 1.07 (s, 3H), 1.00 (s, 3H). ^13^C NMR (75 MHz, CDCl_3_) δ: 164.94, 157.22, 141.69, 139.76, 139.02, 136.17, 134.04, 133.92, 131.17, 130.29, 129.79, 129.07, 128.56, 128.47, 127.41, 127.06, 126.85, 126.67, 124.64, 121.86, 120.85, 75.19, 71.82, 29.17, 26.59, 25.69, 23.03. HR-MS (ESI) *m/z*: Calcd. for C_33_H_32_ClNO_3_ [M+H]^+^, 526.2143; Found, 526.2138.

(6*S*,2*E*)-*N*-(2-(4-Chlorophenyl)phenyl)-3-(4-phenylphenyl)-6,7-dihydroxy-7-methyloct-2-enamide (***S***)-**11l**. A colorless liquid, 275 mg, yield 80%. [α${]}_{D}^{20}$ = +101.9 (c 1.7, CHCl_3_), ee 94.6%. ^1^H NMR (300 MHz, CDCl_3_) δ: 8.37 (d, *J* = 8.1 Hz, 1H), 7.65–7.23 (m, 17H), 5.95 (s, 1H), 5.22 (s, 1H), 3.75–3.65 (m, 1H), 3.37 (d, *J* = 9.2 Hz, 1H), 2.89–2.80 (m, 2H), 1.62–1.49 (m, 2H), 1.07 (s, 3H), 1.00 (s, 3H). ^13^C NMR (75 MHz, CDCl_3_) δ: 164.95, 157.20, 141.68, 139.76, 139.03, 136.18, 134.04, 133.92, 131.21, 130.29, 129.80, 129.07, 128.56, 128.46, 127.41, 127.06, 126.86, 126.67, 124.66, 121.91, 120.85, 75.20, 71.82, 29.18, 26.59, 25.69, 23.04. HR-MS (ESI) *m/z*: Calcd. for C_33_H_32_ClNO_3_ [M+H]^+^, 526.2143; Found, 526.2138.

(6*R*,2*E*)-*N*-(2-chloro-4-trifluoromethylphenyl)-6,7-dihydroxy-7-methyl-3-(naphthalen-1-yl)oct-2-enamide (***R***)-**11m**. A colorless liquid, 97 mg, yield 75%. [α${]}_{D}^{20}$ = −99.8 (c 1.5, CHCl_3_), ee 93.2%. ^1^H NMR (300 MHz, CDCl_3_) δ: 8.71 (d, *J* = 8.7 Hz, 1H), 7.99–7.85 (m, 4H), 7.66 (s, 1H), 7.56–7.45 (m, 4H), 7.30 (d, *J* = 6.8 Hz, 1H), 6.19 (s, 1H), 4.88 (d, *J* = 3.4 Hz, 1H), 3.80–3.72 (m, 1H), 3.65–3.54 (m, 1H), 2.98–2.90 (m, 1H), 2.85 (s, 1H), 1.56–1.49 (m, 2H), 1.08 (s, 3H), 1.07 (s, 3H). ^13^C NMR (75 MHz, CDCl_3_) δ: 164.70, 160.49, 139.09, 137.11, 133.53, 130.16, 128.43, 128.34, 126.27, 125.92, 124.71, 124.68, 124.48, 123.24, 122.39, 121.01, 76.27, 75.61, 72.00, 30.03, 29.28, 25.76, 23.05. HR-MS (ESI) *m/z*: Calcd. for C_26_H_25_ClF_3_NO_3_ [M+H]^+^, 492.1548; Found, 492.1547.

(6*S*,2*E*)-*N*-(2-Chloro-4-trifluoromethylphenyl)-3-(naphthalen-1-yl)-6,7-dihydroxy-7-methyloct-2-enamide (***S***)-**11m**. A colorless liquid, 110 mg, yield 85%. [α${]}_{D}^{20}$ = +99.8 (c 1.7, CHCl_3_), ee 99.4%. ^1^H NMR (300 MHz, CDCl_3_) δ: 8.71 (d, *J* = 8.7 Hz, 1H), 7.99–7.87 (m, 4H), 7.66 (s, 1H), 7.55–7.45 (m, 4H), 7.30 (d, *J* = 6.8 Hz, 1H), 6.19 (s, 1H), 4.88 (d, *J* = 3.4 Hz, 1H), 3.80–3.72 (m, 1H), 3.65–3.54 (m, 1H), 2.98–2.90 (m, 1H), 2.84 (s, 1H), 1.56–1.50 (m, 2H), 1.07 (s, 3H), 1.06 (s, 3H). ^13^C NMR (75 MHz, CDCl_3_) δ: 164.70, 160.49, 139.10, 137.12, 133.53, 130.16, 128.42, 128.34, 126.27, 125.91, 124.71, 124.68, 124.48, 123.24, 122.40, 121.03, 75.63, 72.00, 30.04, 29.21, 25.76, 23.06. HR-MS (ESI) *m/z*: Calcd. for C_26_H_25_ClF_3_NO_3_ [M+H]^+^, 492.1548; Found, 492.1547.

(6*R*,2*E*)-*N*-(2-(4-Chlorophenyl)phenyl)-3-(naphthalen-1-yl)-6,7-dihydroxy-7-methyloct-2-enamide (***R***)-**11n**. A colorless liquid, 116 mg, yield 94%. [α${]}_{D}^{20}$ = −107.9 (c 1.6, CHCl_3_), ee 96.2%. ^1^H NMR (300 MHz, CDCl_3_) δ: 8.40 (d, *J* = 8.2 Hz, 1H), 7.94–7.80 (m, 3H), 7.54–7.38 (m, 6H), 7.34–7.19 (m, 6H), 5.85 (s, 1H), 5.30 (d, *J* = 2.9 Hz, 1H), 3.85–3.75 (m, 1H), 3.59 (d, *J* = 9.8 Hz, 1H), 2.94 (s, 1H), 2.88–2.77 (m, 1H), 1.53–1.40 (m, 2H), 1.06 (s, 3H), 1.05 (s, 3H). ^13^C NMR (75 MHz, CDCl_3_) δ: 164.69, 157.91, 139.26, 136.06, 133.98, 133.93, 133.48, 131.34, 130.22, 129.77, 129.05, 128.49, 128.28, 128.16, 126.11, 125.82, 124.77, 124.65, 124.51, 123.82, 121.92, 75.24, 71.91, 29.68, 29.00, 25.76, 23.00. HR-MS (ESI) *m/z*: Calcd. for C_31_H_30_ClNO_3_ [M+H]^+^, 500.1987; Found, 500.1985.

(6*S*,2*E*)-*N*-(2-(4-Chlorophenyl)phenyl)-3-(naphthalen-1-yl)-6,7-dihydroxy-7-methyloct-2-enamide (***S***)-**11n**. A colorless liquid, 114 mg, yield 92%. [α${]}_{D}^{20}$ = +109.1 (c 2.0, CHCl_3_), ee 93.2%. ^1^H NMR (300 MHz, CDCl_3_) δ: 8.40 (d, *J* = 8.2 Hz, 1H), 7.94–7.80 (m, 3H), 7.54–7.38 (m, 6H), 7.34–7.19 (m, 6H), 5.85 (s, 1H), 5.30 (d, *J* = 2.9 Hz, 1H), 3.85–3.75 (m, 1H), 3.59 (d, *J* = 9.8 Hz, 1H), 2.94 (s, 1H), 2.88–2.78 (m, 1H), 1.53–1.40 (m, 2H), 1.06 (s, 3H), 1.05 (s, 3H). ^13^C NMR (75 MHz, CDCl_3_) δ: 164.70, 157.90, 139.26, 136.07, 133.98, 133.92, 133.48, 131.36, 130.23, 129.78, 129.05, 128.49, 128.28, 128.16, 126.11, 125.83, 124.77, 124.65, 124.52, 123.83, 121.94, 75.24, 71.92, 29.68, 29.00, 25.76, 23.00. HR-MS (ESI) *m/z*: Calcd. for C_31_H_30_ClNO_3_ [M+H]^+^, 500.1987; Found, 500.1985.

(6*R*,2*E*)-*N*-(2-Chloro-4-trifluoromethylphenyl)-3-(naphthalen-2-yl)-6,7-dihydroxy-7-methyloct-2-enamide (***R***)-**11o**. A colorless liquid, 62 mg, yield 78%. [α${]}_{D}^{20}$ = −102.1 (c 0.5, CHCl_3_), ee 95.2%. ^1^H NMR (300 MHz, CDCl_3_) δ: 8.69 (d, *J* = 8.7 Hz, 1H), 7.99 (s, 1H), 7.95–7.84 (m, 4H), 7.67 (s, 1H), 7.60–7.50 (m, 4H), 6.34 (s, 1H), 4.83 (d, *J* = 3.6 Hz, 1H), 3.74 (ddd, *J* = 13.6, 10.4, 6.4 Hz, 1H), 3.47–3.40 (m, 1H), 3.09–3.04 (m, 1H), 2.71 (s, 1H), 1.63–1.55 (m, 2H), 1.05 (s, 3H), 0.98 (s, 3H). ^13^C NMR (75 MHz, CDCl_3_) δ: 164.98, 159.97, 137.28, 137.18, 133.37, 132.86, 128.39, 128.08, 127.37, 126.67, 126.46, 126.04, 125.92 (q, *J* =3.9 Hz), 124.65 (q, *J* =3.3 Hz), 123.81, 122.35, 121.01, 120.67, 76.24, 75.58, 71.88, 29.50, 26.96, 25.67, 23.05. HR-MS (ESI) *m/z*: Calcd. for C_26_H_25_ClF_3_NO_3_ [M+H]^+^, 492.1548; Found, 492.1546.

(6*S*,2*E*)-*N*-(2-Chloro-4-trifluoromethylphenyl)-3-(naphthalen-2-yl)-6,7-dihydroxy-7-methyloct-2-enamide (***S***)-**11o**. A colorless liquid, 89 mg, yield 83%. [α${]}_{D}^{20}$ = +102.5 (c 1.3, CHCl_3_), ee 93.4%. ^1^H NMR (300 MHz, CDCl_3_) δ: 8.68 (d, *J* = 8.7 Hz, 1H), 8.00 (s, 1H), 7.95–7.84 (m, 4H), 7.67 (s, 1H), 7.60–7.50 (m, 4H), 6.35 (s, 1H), 4.84 (d, *J* = 3.6 Hz, 1H), 3.74 (ddd, *J* = 13.6, 10.4, 6.4 Hz, 1H), 3.47–3.40 (m, 1H), 3.09–3.04 (m, 1H), 2.74 (s, 1H), 1.63–1.55 (m, 2H), 1.05 (s, 3H), 0.98 (s, 3H). ^13^C NMR (75 MHz, CDCl_3_) δ: 164.99, 159.93, 137.29, 137.20, 133.37, 132.86, 128.38, 128.09, 127.37, 126.67, 126.46, 126.04, 125.92 (q, *J* = 3.8 Hz), 124.64 (q, 4.4 Hz), 123.82, 122.40, 121.06, 120.66, 76.26, 75.62, 71.90, 29.52, 26.97, 25.67, 23.07. HR-MS (ESI) *m/z*: Calcd. for C_26_H_25_ClF_3_NO_3_ [M+H]^+^, 492.1548; Found, 492.1546.

(6*R*,2*E*)-*N*-(2-(4-Chlorophenyl)phenyl)-3-(naphthalen-2-yl)-6,7-dihydroxy-7-methyloct-2-enamide (***R***)-**11p**. A colorless liquid, 110 mg, yield 85%. [α${]}_{D}^{20}$ = −97.4 (c 1.9, CHCl_3_), ee 97.8%. ^1^H NMR (300 MHz, CDCl_3_) δ: 8.37 (d, *J* = 8.1 Hz, 1H), 7.86–7.83 (m, 4H), 7.55–7.23 (m, 11H), 6.02 (s, 1H), 5.26 (d, *J* = 3.0 Hz, 1H), 3.80–3.64 (m, 1H), 3.43–3.38 (m, 1H), 3.00–2.90 (m, 1H), 2.85 (s, 1H), 1.62–1.44 (m, 2H), 1.03 (s, 3H), 0.94 (s, 3H). ^13^C NMR (75 MHz, CDCl_3_) δ: 165.00, 157.52, 137.56, 136.21, 134.03, 133.89, 133.19, 132.84, 131.36, 130.28, 129.83, 129.05, 128.46, 128.22, 128.00, 127.33, 126.46, 126.35, 125.86, 124.72, 124.00, 122.06, 121.52, 75.22, 71.82, 29.17, 26.70, 25.67, 22.98. HR-MS (ESI) *m/z*: Calcd. for C_31_H_30_ClNO_3_ [M+H]^+^, 500.1987; Found, 500.1987.

(6*S*,2*E*)-*N*-(2-(4-Chlorophenyl)phenyl)-3-(naphthalen-2-yl)-6,7-dihydroxy-7-methyloct-2-enamide (***S***)-**11p**. A colorless liquid, 100 mg, yield 78%. [α${]}_{D}^{20}$ = +98.6 (c 1.8, CHCl_3_), ee 94.2%. ^1^H NMR (300 MHz, CDCl_3_) δ: 8.37 (d, *J* = 8.1 Hz, 1H), 7.86–7.83 (m, 4H), 7.55–7.23 (m, 11H), 6.02 (s, 1H), 5.26 (d, *J* = 3.0 Hz, 1H), 3.80–3.67 (m, 1H), 3.43–3.38 (m, 1H), 3.00–2.90 (m, 1H), 2.85 (s, 1H), 1.62–1.45 (m, 2H), 1.03 (s, 3H), 0.94 (s, 3H). ^13^C NMR (75 MHz, CDCl_3_) δ: 165.02, 157.50, 137.56, 136.23, 134.04, 133.88, 133.19, 132.84, 131.38, 130.28, 129.84, 129.04, 128.45, 128.22, 128.00, 127.33, 126.46, 126.35, 125.86, 124.73, 124.01, 122.10, 121.51, 75.23, 71.82, 29.17, 26.71, 25.67, 22.99. HR-MS (ESI) *m/z*: Calcd. for C_31_H_30_ClNO_3_ [M+H]^+^, 500.1987; Found, 500.1987.

### 3.6. Fungicidal Activity of the Amides ***11a***–***11p***

The in vitro fungicidal activities of compounds **11a**–**11p** against *R. Solani*, *A. Solani*, *F. graminearum*, *S. Sclerotiorum*, *B. cinerea* and *P. capsici* were evaluated using methods in the references [27,28] by the mycelium growth rate. Procedure for inhibition rate: The stock 2000 µg/mL DMSO solutions of tested compounds were prepared in advance. Then hot potato dextrose agar (PDA) culture medium (9.75 mL) was added into a plate, and we added sample solution (0.25 mL) or blank DMSO (0.25 mL) to the plate and mixed with PDA culture medium, to make the final concentration 50 µg/mL. When the plate was made, we put a 5 mm diameter fungus cake into the center of plate, incubated them at 25 ± 0.5 °C for 24–48 h, checked the growth status and calculated the inhibition rate according to the reference. Three replicates were performed and the mean measurements were calculated from the three replicates for each concentration. The EC_50_ values were determined from the inhibition rates of six different concentrations (100, 25.0, 6.25, 1.56, 0.39, 0.10 µg/mL) based on the statistics method for the compounds which had more than 70% inhibition rates in the preliminary evaluation. Dimethomorph and pyrimorph were used as the positive control in the mycelium growth rate test.

The in vivo fungicidal activities of compounds **11a**–**11p** against *Pseudoperonospora cubensis*, *Erysiphe graminis*, *Puccinia sorghi* and *Colletotrichum gloeosporioides* were evaluated using the potted plant method in a greenhouse [29,30]. Flumorph and pyrimorph were used as the positive control. The evaluation experiments were performed by State Key Laboratory of the Discovery and Development of Novel Pesticide, Shenyang Sinochem Agrochemicals R&D Co. Ltd., Shenyang, China.

## 4. Conclusions

In conclusion, novel cinnamide fungicidal leads with optical hydroxyl side chain (*R*)-**11a**–**11p** and (*S*)-**11a**–**11p** were designed and synthesized through amidation of the olefin acids **7** and **8a**–**8d** or Mizoroki–Heck arylation of the amides **9a** and **9b**, and stereoselective synthesis of optical isomers of 3-aryl-7-methyl-6,7-dihydroxyoct-2-enamide with Sharpless asymmetric dihydroxylation as the key steps. Their structures were characterized by the ^1^H, ^13^C NMR and HR-ESI-MS spectra data, and the e.e values were analyzed by chiral HPLC. The EC_50_ values of (*R*)-**11f**, (*R*)-**11m**, (*S*)-**11m** and (*R*)-**11n** were 0.16, 0.28, 0.41 and 0.47 µg/mL against *S. sclerotiorum* in the in vitro evaluation, respectively. The efficacies of (*R*)- and (*S*)-**11i** and **11j** against *P. cubensis* in the in vivo evaluation were 100% at 400 µg/mL, which showed they were the most active compounds and could be used as the potential lead structures for the further modification. 

## Data Availability

Not applicable.

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
