# Peer review of "Discovery of Novel Cinnamide Fungicidal Leads with Optical Hydroxyl Side Chain"

_molecules, 2022, doi:10.3390/molecules27165259_

Round 1

Reviewer 1 Report

A relevant issue underlying food security, as loss of yield and crops quality are due to the resistance of phytopathogens.

 However, here are some issues that should be addressed/corrected/considered:

 1 – Rewrite the third sentence of the Introduction chapter (lines 27-28) to make it more correct.

In any case, the beginning of the Introduction chapter is interestingly written, opening the way for the discovery of the challenges mentioned by the authors.

 2 – Although presented already in the literature, a concise summary of the resistance (and resistant mechanisms) to most relevant fungicides, and the compounds mostly used in the Chinese market, would be very useful, as a bridge to the need for developing novel chemical structures. The cost of this development and the difficulties of novel chemical structures could also be addressed.

 3 – Scheme 2 must be addressed in the text when describing compound 1 (Line 38) for the first time.

 4 – (Line 45): which reference is the previous paper?

 5 – Inhibition rates (%) must be accompanied by standard deviation (SD) values. Statistical significance (F, p) must be given.

 6 – Although values of inhibition rates were always higher towards S. sclerotiorum, considering all tested compounds, 0% was found for (R)-11e (R1, R2, morpholino). It would be interesting to discuss these findings.

 7 – Any possible explanation for the high values of in vivo activity considering (R)- and (S)-11i and 11j?

 8 - Bearing in mind that the ultimate goal in the development of new fungicides is the stopping of resistance (and not just the delay of this resistance), how do the authors address this issue in their strategies to use these novel compounds as potential lead structures?

Author Response

please see attached letter.

Reviewer 2 Report

the article is interesting to read however, some comments to the authors to consider:

1. please define abbreviations the first time they appear in the text such as abstract

2. please write the in vivo and in vitro in italic format

3. Would it be possible for the authors to comment on the compounds that contain  cinnamoyl, pyridine ring, and 2-phenylethylamine moieties in pesticides?

4. would it be possible for example to comment on the mode of action for such compounds and how they interfere with the Ergosterol pathway or any other selective pathways

5. would it be possible to determine the PI, PP, PS, and RS comparatively between the compounds reported in the manuscript

6. could the authors expand on the in vivo fungicidal activity of the compounds 11a-11p, please

7.     In the materials section, please list all chemical used and their suppliers. Could you please also provide the methods section solutions, concentrations, instrument settings, and the exact amount added should be stated clearly? Take extra attention to instrumental settings in which the measurement was recorded

Author Response

please see attached letter.
